# Optimization of Storage Conditions of the Medicinal Herb *Ilex asprella* against the Sterigmatocystin Producer *Aspergillus versicolor* Using Response Surface Methodology

**DOI:** 10.3390/toxins10120499

**Published:** 2018-11-27

**Authors:** Xiaofang Lu, Chaoquan Luo, Jianyong Xing, Zhengzhou Han, Tong Li, Weiwei Wu, Hui Xu, Ruoting Zhan, Weiwen Chen

**Affiliations:** 1Research Center of Chinese Herbal Resource Science and Engineering, Guangzhou University of Chinese Medicine, Guangzhou 510006, China; lxf8717@gmail.com (X.L.); 512684691@qq.com (C.L.); ruotingzhan@vip.163.com (R.Z.); chenww@gzucm.edu.cn (W.C.); 2Ministry of Education Key Laboratory of Chinese Medicinal Resource from Lingnan, Guangzhou 510006, China; 3Joint Laboratory of National Engineering Research Center for the Pharmaceutics of Traditional Chinese Medicines, Guangzhou 510006, China; 4China Resources Sanjiu Medical & Pharmaceutical Co, Ltd., Shenzhen 518110, China; xingjianyong@999.com.cn (J.X.); gulf@999.com.cn (Z.H.); 5School Materia Medica, Guangzhou University of Chinese Medicine, Guangzhou 510006, China; litong9801@outlook.com (T.L.); wuweiwei717@outlook.com (W.W.)

**Keywords:** *Ilex asprella*, *Aspergillus versicolor*, sterigmatocystin, response surface methodology, storage conditions, temperature, humidity

## Abstract

The root of *Ilex asprella* is a commonly used herb in Southern China, and also constitutes the main raw material of Canton herbal tea. *I. asprella* is readily contaminated by mildew because of rich nutrients. *Aspergillus versicolor* producing sterigmatocystin is one of the most common molds that contaminate foodstuffs and medicinal herbs. Previous study on the evaluation of fungal contamination on samples of *I. asprella* revealed that *A. versicolor* was the dominant contaminant. In this study, experiments based on response surface methodology combined with central composite design were carried out to determine the optimal storage conditions of *I. asprella* to minimize the contamination of sterigmatocystin. The herb, manually innoculated with *A. versicolor,* was stored under different temperatures (20–40 °C) and humidity (80–95%) for seven days. The effects of temperature and humidity were evaluated using total saponin, polysaccharide and the sterigmatocystin levels as quality indexes. The results showed that *A. versicolor* grew quickly and produced large amounts of sterigmatocystin on *I. asprella*, at humidity ranging from 85% to 90% and temperatures above 26 °C. Meanwhile, total saponin and polysaccharide amounts were reduced significantly. These findings suggested that *I. asprella* samples should be stored in an environment with humidity and temperature below 85% and 26 °C, respectively, to reduce *A. versicolor* growth and sterigmatocystin production.

## 1. Introduction

Toxigenic fungi are widely distributed in nature, and not only pollute crops, but also contaminate medicinal plants [1,2]. Halt et al. [3] performed a pollution survey of 62 medicinal plants in Croatia, of which 18% were contaminated by toxigenic fungi.

*Aspergillus versicolor* producing sterigmatocystin (ST) is one of the most common mildews that contaminate foodstuffs and medicinal herbs. ST has obvious carcinogenicity and mutagenicity for animals and humans, with toxicity only inferior to aflatoxin [4]. In addition, ST is an intermediate product of aflatoxin biosynthesis [5]. The International Cancer Research Institute (IARC) classifies ST as a Class 2B carcinogen. ST is also produced by different *Aspergillus* species as well as other species such as *Bipolaris*, *Chaetomium* and *Emiricella* [6]. Vesonder et al. reported that multiple dairy cattle had bloody flux, which reduced milk production, with some animals dying; this was confirmed to be caused by ST in animal feed [7]. In the YinNa areas of Northwest China, a flock of sheep was reported to suffer from the so-called “yellow” disease for a long time; examination revealed that the animals were fed mildew contaminated grass in winter, with ST content as high as 6.5 mg/kg [8]. Recently, Zheng et al. [9] assessed 365 samples of medicinal herbs collected in Guangzhou, China, and ST was detected in one fourth of them.

*Ilex asprella* is one of the most popular medicinal herbs in South China. The root of *I. asprella* is rich in triterpenoid saponins, and has various pharmacological effects; it has been used to treat wind–heat cold, chronic pharyngitis and sore throat [10]. As a “drug homologous food”, it is also included in many herbal teas. *I. asprella* is highly susceptible to mildew, which might result from the extra large amount of polysaccharides it contains, with a content of up to 10% [11]. In our previous study, fungi on the surface of 22 commercial *I. asprella* samples were isolated, purified and identified by ITS sequencing as reported by Bellemain et al. [12]. In total, 29 fungal strains were isolated, including six proven to be ST producers as determined by LC-MS/MS. The above findings implied that the contamination risk of ST producers for *I. asprella* can be quite high and should not be overlooked.

To minimize fungal infection and mycotoxin accumulation, it is necessary to store medicinal herbs and foodstuffs in appropriate conditions. Based on previous reports, temperature and humidity are two major environmental factors [13,14]. Therefore, in the present study, manually inoculated *I. asprella* samples were stored under various temperature and humidity conditions, designed by response surface methodology combined with central composite design (RSM-CCD), and the sample quality was evaluated to provide clues for optimal storage conditions against ST producing fungi. The standard strain *A. versicolor* was selected as a representative strain to inoculate samples. Polysaccharides, total saponins and sterigmatocystin (ST) were used as quality indexes.

## 2. Results and Discussion

### 2.1. Growth of A. versicolor after Manual Inoculation

After inoculation with *A. versicolor* suspension, *I. asprella* samples were incubated under specific temperature and humidity conditions for seven days. *A. versicolor* growth varied greatly among different test groups, and luxuriant growth was observed at the temperature of 30 °C and relative humidity of 88%. Typical samples observed under the sunlight are shown in Figure 1.

### 2.2. Sterigmatocystin Content in I. asprella

After incubation for seven days, ST levels in *I. asprella* samples were determined by a HPLC-MS/MS method developed previously [9]. Typical SRM chromatograms are shown in Figure 2. ST contents ranged from 8.4 to 19.6 μg/kg (Table 1).

### 2.3. Polysaccharide and Total Saponin Contents

To quantify the polysaccharide content of *I. asprella*, the phenol–sulfuric acid method was used. The calibration curve generated with glucose standard was linear within the range of 5–50 μg/mL, with a correlation coefficient of 0.99. The average polysaccharide content of *I. asprella* was 12.71%. After incubation with *A. versicolor* for seven days, the polysaccharide content was reduced remarkably (Table 1), especially for samples stored at 30 °C and 88% humidity. Furthermore, under these conditions, *A. versicolor* grew well, with very high ST concentrations. To assess total saponin amounts in *I. asprella*, colorimetry detecting vanillin-glacial acetic acid was applied. The calibration curve generated with ursolic acid was linear within the range of 25–400 μg/mL, with a correlation coefficient of 0.99. The average content of total saponins was 10.83 mg/g. Similar to polysaccharide, total saponin amounts were reduced remarkably after incubation, as shown in Table 1.

### 2.4. Model Fitting and Statistical Analysis

The experiments showed that all variables examined in this study had effects on ST, total saponin amounts and polysaccharide production. Therefore, the effects of two variables, including temperature and humidity on responses (levels of ST, total saponins and polysaccharides) were examined using central composite design (CCD). Table 1 depicts the complete design matrix as well as response values. Multiple regression analysis provided predicted responses (Y values) for ST, total saponins and polysaccharides, via second-order polynomial Equations (1)–(3) as follows:ST: *Y* = −0.057317 × *T*^2^ − 0.11841 × *H*^2^ + 0.030847 × *T* × *H* + 0.66258 × *T* + 19.89322 × *H* − 865.79897(1)
Total saponins: *Y* = 0.028156 × *T*^2^ + 0.023946 × *H*^2^ + 0.00739582 × *T* × *H* − 2.46118 × *T* − 4.59722 × *H* + 250.82892(2)
Polysaccharides: *Y* = 0.049183 × *T*^2^ + 0.036444 × *H*^2^ − 0.015132 × *T* × *H* − 1.64457 × *T* − 6.19986 × *H* + 313.28930(3)
where *Y* is the predicted response (levels of ST, total saponins and polysaccharides, respectively); *T* and *H* represent two independent variables, temperature and humidity, respectively.

Table 2 summarizes the data for analysis of variance by Fisher’s *F* test, the goodness-of-fit and the adequacy of all models. For ST, a quadratic regression model with high significance was found with high *F* and extremely low *p*- (*p* = 0.0065) values, suggesting the combined impacts of different independent variables markedly contributed to response maximization [15]. Meanwhile, there was lack of fit (*p* = 0.3397) confirming that these data fit well with this model. A determination coefficient (*R*^2^) of 0.8963 indicated that a variation of 89.63% for ST production was due to independent parameters in samples. Equation (1) equally had an adjusted correlation coefficient (*R*^2^_adj_) of 0.8099, which satisfactorily confirmed the model’s significance [16]. The higher the *R*^2^_adj_ value, the more actual and predicted values are correlated [17]. For Equation (2), (*p* < 0.05) and an *F*-value of 11.55 indicated a useful model. An *R*^2^ of 0.9060 suggested about 9.4% of the overall variance could not be due to the response. *R*^2^_adj_ was 0.8120, suggesting a reliable mathematical model, which was suitable for determining total saponin amounts. For Equation (3), *p* < 0.05 and an *F*-value of 9.51 indicated a useful model. *R*^2^ of 0.8879 was derived, for an *R*^2^_adj_ of 0.7945, indicating the mathematical model had good reliability, and could be applied to assess polysaccharide amounts. Moreover, *p* values of humidity were higher than those of temperature in all three models, indicating that the former demonstrated more significant influence on *I. asprella* quality than the latter.

### 2.5. Response Surface Analysis

The effects of the independent parameters, temperature and humidity, as well as their interactive impacts on the amounts of ST, total saponins and polysaccharides were depicted by 2-dimensional contour plots and 3-dimensional response surface plots, respectively, employing RSM. Regression models in Equations (1)–(3) helped predict responses, deriving ultimate values for ST, total saponins and polysaccharide levels [18,19].

As shown in Figure 3, temperatures and humidity levels ranging from 26 °C to 34 °C and 85% to 90%, respectively, significantly increased ST content, indicating that these conditions were optimal for *A. versicolor* to produce ST. Based on the data in Table 1, the temperature and humidity for the maximum production of ST were calculated to be 30 °C and 88%, respectively. In comparison, increasing temperature ranging from 26 °C to 37 °C and relative humidity of 85–95% reduced the amounts of total saponins and polysaccharides remarkably (Figure 4 and Figure 5). Therefore, to ensure optimal chemical composition for the herb as well as minimize ST-hazard risk, the recommended storage conditions would be humidity below 85% and temperature below 26 °C.

## 3. Conclusions

Stored *I. asprella* is an artificial ecosystem with qualitative and nutritive changes resulting from the interactions of physical, chemical and biological factors. Fungal damage and mycotoxin contamination constitute major concerns. Under satisfactory storage conditions, mycotoxin contamination can occur [20,21]. Elevated temperature and relative humidity represent critical parameters promoting mold growth and mycotoxin contamination during cereal, feed and herb storage [22]. *I. asprella* represents a commonly cultivated plant in tropical and subtropical areas, especially in Chinese Guangdong province, with a warm climate, heavy rainfall and high humidity. Therefore, *I. asprella* root might show elevated sensitivity to mildew during storage in these conditions.

Response surface methodology allows the determination of an optimal response based on a few sets of experiments, in which all factors vary within a chosen range. Conclusively, the proposed model has shown to be significantly important in statistical terms. This study showed that *A. versicolor* growth and toxicity were affected by environmental temperature and humidity. The effects of humidity on growth, mycotoxin production and the alteration of effective components were higher than those of temperature. Samples of *I. asprella* should be conserved under low temperature 26 °C and humidity 85%, to reduce ST content, while maintaining the amounts of active ingredients. However, concerning the storage time, no conclusion could be drawn from this study. A long-term experiment with real samples might be necessary to figure out the answer. This study showed that it was scientific and feasible to use the RSD-CCD method to assess the conditions of herb storage, and provided new clues for controlling the pollution of medicinal plants by mycotoxin.

## 4. Materials and Methods

### 4.1. Chemicals and Reagents

Mycotoxin standard (ST) was purchased from Supelco Sigma Aldrich (St. Louis, MO, USA), dissolved in methanol (1 µg/mL) and stored at −20 °C until analysis. *A. versicolor* strain As3.4413 was purchased from China General Microbiological Culture Collection Center (Beijing, China). Organic solvents, including methanol, acetonitrile and formic acid, were of HPLC grade and supplied by Merck (Merck KGaA, Germany). All the other organic reagents were of analytical grade and purchased from Tianjin Chemical Reagent (Tianjin, China). Pure water was obtained from a Milli-Q system (Millipore, Billerica, MA, USA).

### 4.2. Plant Material Preparation and Pretreatment

*I. asprella* roots were collected from fields, sliced and dried immediately. The resulting samples were stored in sealed bags at −20 °C until use. The samples (10g each) were sterilized by ultraviolet radiation for 1 h. Then, 1.0 mL spore suspension of *A. versicolor* (10^6^ spores/mL) was added accurately to the dish. According to the conditions described in Table 1, different storage conditions were simulated for seven days.

### 4.3. Experimental Design

In this study, RSM was performed to assess the associations of mycotoxin, total saponin and polysaccharide production. A two-factor, three-level design was developed by CCD to generate a second-order polynomial model. Then, two variables, namely temperature (A) and humidity (B), and their appropriate ranges were determined based on single-factor experiments.

The predicted and actual levels of the independent variables are presented in Table 3.

### 4.4. Determination of Sterigmatocystin by HPLC-MS/MS

Mycotoxin extraction from the plant was performed as previously described by our team [9]. Briefly, for extraction, 2.0 g of powder was diluted in 10 mL acetonitrile-water (84/16, *v*/*v*). The sample was mixed for 60 min at maximum speed on a Vortex. The solution underwent centrifugation at 5000 rpm for 5 min at room temperature. Afterward, the supernatant was decanted and filtered through a membrane with a 0.22 μm Millipore filter. HPLC–MS/MS was used for ST purification and quantitation, using a Thermo Fisher Scientific Hypersil GOLD C_18_ column (2.1 mm × 100 mm, 3 μm). The mobile phase, composed of methanol (A) and 4.0 mM ammonium acetate 0.1% formic acid aqueous solution (B), was programmed as follows: gradient elution (0–5 min, 20% A; 5–20 min, 20% A–80% A; 20–30 min, 80% A–100% A; 30–35 min, 100% A). The flow rate was 0.3 mL/min and an injection volume of 10 μL was adopted, with a column temperature at 30 °C. Identification of the isolated mycotoxin was performed by HPLC–MS/MS in the HESI (+) mode and analytical HPLC by comparing retention times and MS data with analytical standards. For mass detection, the precursor (*m*/*z* = 325), quantitative (*m*/*z* = 281) and qualitative (*m*/*z* = 310) were used for ST analysis. ST detection of relevant mass spectrum parameters is shown in Table 4.

### 4.5. Determination of Total Saponin Content

The ursolic acid standard was diluted to a concentration 1 mg/mL. For extraction, 1.0 g of powder was diluted in 20 mL of ethanol/water (70:30 *v*/*v*), followed by ultrasonication for 30 min. Filtration after cooling was performed using ethanol/water at constant volume of 50 mL, and the solution underwent centrifugation at 8000 rpm for 10 min at room temperature. Afterward, 5 mL of the supernatant was mixed well with 5 mL of hydrochloric acid (7:93, *v*/*v*). The mixture was placed in a water bath at 90 °C for 30 min, followed by ethyl acetate extraction 3 times. Then, 0.4 mL of the extract was placed in a 10 mL colorimetric tube and dried; after the addition of 0.4 mL of vanillin-glacial acetic acid (5:95, *v*/*v*) and 1.6 mL of perchloric acid, the sample was placed in a water bath at 60 °C for 15 min and mixed with 8 ml of glacial acetic acid. Finally, absorbance was measured at 545 nm, and total saponin levels were calculated based on a standard curve generated with ursolic acid.

### 4.6. Determination of Polysaccharide Amounts

Polysaccharides were extracted from *I. asprella* and quantified as described previously [11]. One gram of *I. asprella* powder was diluted in 50 mL of water, followed by ultrasonication for 60 min; the sample was diluted to 200 mL after filtration. Then, 2 mL of the solution was mixed with 1 mL phenol and 5 mL vitriol, and placed 10 min before incubation in a water bath at 40 °C for 15 min. Absorbance was measured at 490 nm on an ultraviolet spectrophotometer. The polysaccharide content was calculated based on a standard curve generated with glucose.

### 4.7. Statistical Analysis

Stat-Ease software (Design-Expert 8.0.6.1 version, Stat-Ease Corporation, Minneapolis, MN, USA, 2015) was used for the regression analysis of the data, and to plot the response surface graphs. The variability and accuracy of the model were determined according to the regression coefficient (*R*^2^) and lack of fit, respectively. Analysis of variance (ANOVA) was performed and the values were considered significant when *p* < 0.05. The response surface and contour plots of the predicted responses of the model were used to assess interactions between the significant factors. Additionally, numerical optimization was carried out by performing three-dimensional response surface analysis of the independent and dependent variables.

## Figures and Tables

**Figure 1 toxins-10-00499-f001:**
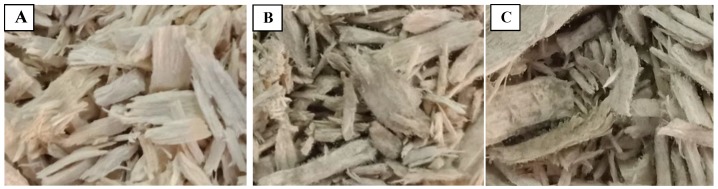
Typical samples observed under the sunlight. (**A**) negative control; (**B**) inoculated sample after incubation for seven days at 23 °C and 82% humidity; (**C**) inoculated sample after incubation for seven days at 30 °C and 88% humidity.

**Figure 2 toxins-10-00499-f002:**
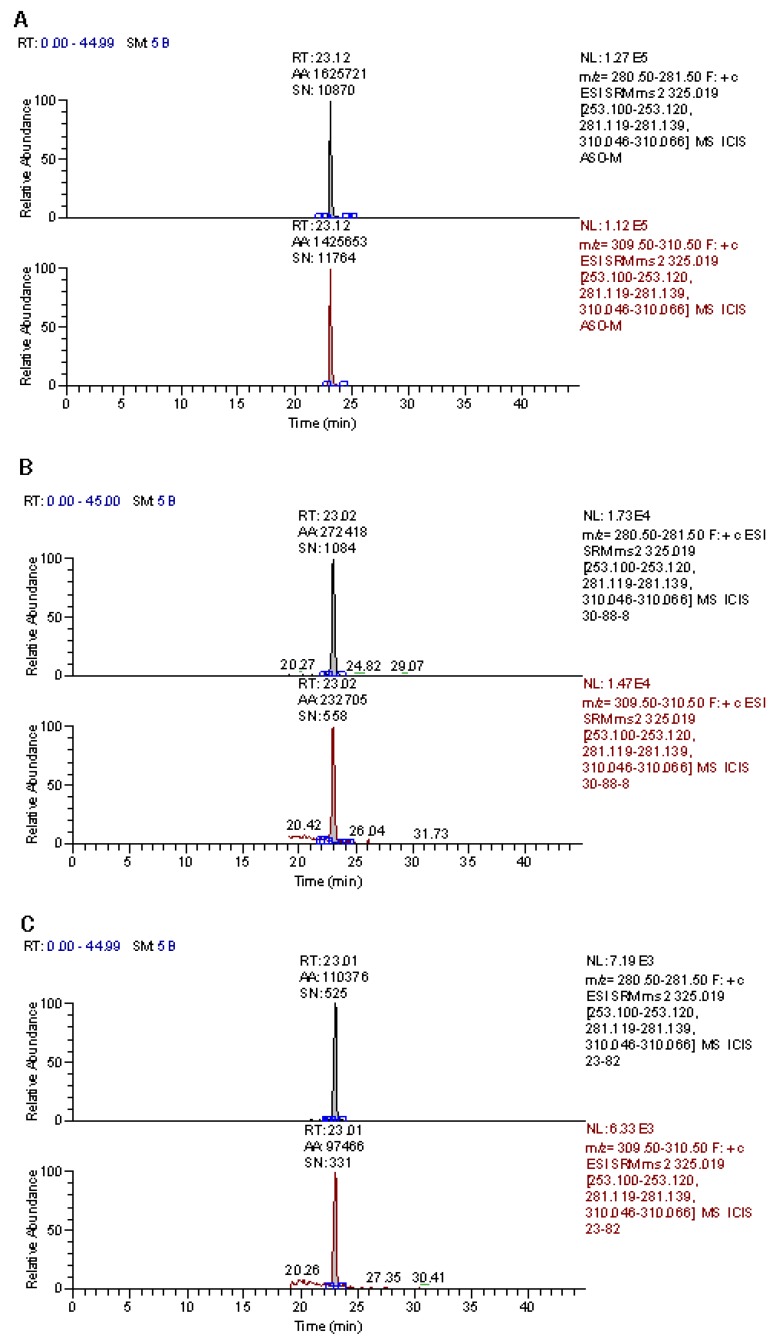
Typical SRM chromatograms of sterigmatocystin (ST) containing *I. asprella* samples. (**A**) ST standard; (**B**) inoculated sample after incubation for seven days at 30 °C and 88% humidity; (**C**) inoculated sample after incubation for seven days at 23 °C and 82% humidity. Top, quantitative ion; bottom, qualitative ion.

**Figure 3 toxins-10-00499-f003:**
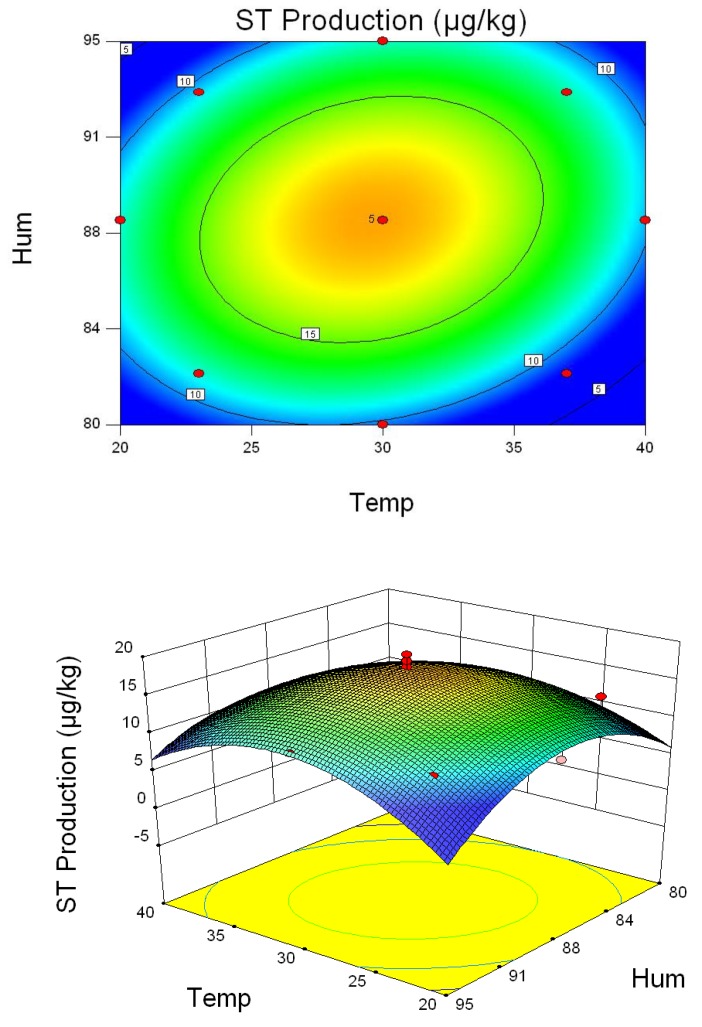
Three-dimensional response surface plots and corresponding contour plots of parameters for ST content in *A. versicolor*. The contact between temperature and humidity is shown.

**Figure 4 toxins-10-00499-f004:**
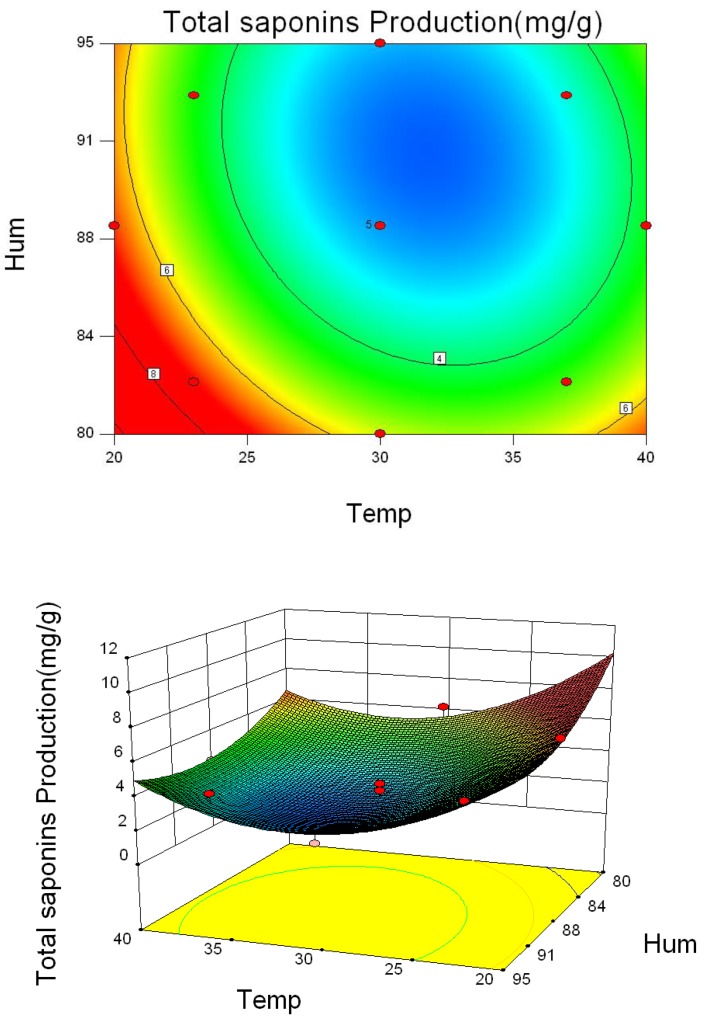
Three-dimensional response surface plots and corresponding contour plots of parameters for total saponin content in *A. versicolor*. The contact between temperature and humidity is shown.

**Figure 5 toxins-10-00499-f005:**
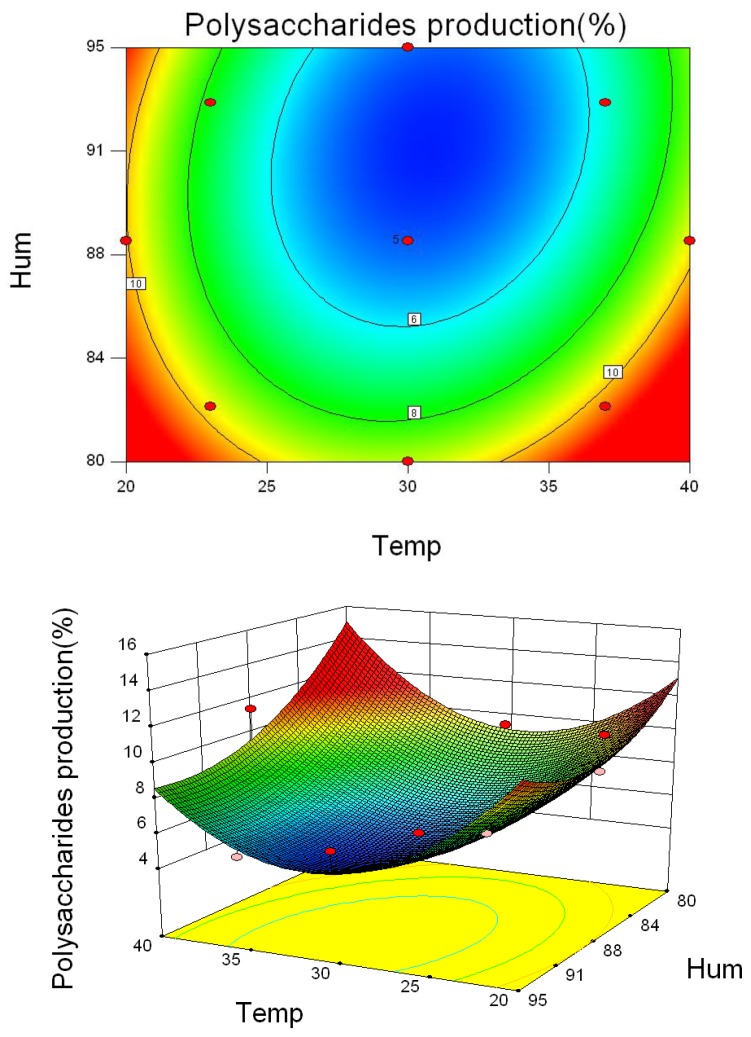
Three-dimensional response surface plots and corresponding contour plots of parameters for polysaccharide content in *A. versicolor*. The contact between temperature and humidity is shown.

**Table 1 toxins-10-00499-t001:** Outline of experimental design and the responses observed in *I. asprella* samples.

Test	Temp.	Hum.	ST (μg/kg)	Total Saponins (mg/g)	Polysaccharides (%)
1	23	82	13.0	6.48	9.93
2	37	82	8.8	4.47	9.45
3	23	93	11.2	4.67	7.48
4	37	93	11.6	3.86	4.39
5	20	88	10.6	7.07	9.73
6	40	88	10.6	4.12	11.38
7	30	80	8.4	6.21	9.4
8	30	95	12.2	2.25	6.28
9	30	88	18.0	3.52	5.03
10	30	88	19.6	1.83	4.41
11	30	88	18.8	3.1	4.66
12	30	88	17.2	2.5	4.28
13	30	88	16.8	2.17	4.72

**Table 2 toxins-10-00499-t002:** Response surface quadratic models given by ANOVA for assessing target mycotoxin, total saponin amounts and polysaccharide production.

Source	Sum of Squares	df	Mean Square	*F*-Value	*p*-Value
**ST**					
Model	74.2	5	14.84	10.37	0.0065
Temp	2.22	1	2.22	1.55	0.2597
Hum	1.85	1	1.85	1.29	0.2993
TH	5.68	1	5.68	3.97	0.0935
T^2^	25.35	1	25.35	17.72	0.0056
H^2^	62.73	1	62.73	43.84	0.0006
Residual	8.59	6	1.43		
Lack of fit	5.39	3	1.80	1.68	0.3397
Pure Error	3.20	3	1.07		
Cor Total	185.76	12			
**Total saponins**					
Model	29.39	5	5.88	11.55	0.0049
Temp	6.13	1	6.13	12.05	0.0133
Hum	7.97	1	7.97	15.67	0.0075
TH	0.33	1	0.33	0.64	0.4542
T^2^	12.98	1	12.98	25.51	0.0023
H^2^	3.10	1	3.10	6.09	0.0486
Residual	3.05	6	0.51		
Lack of fit	2.17	3	0.72	2.48	0.2379
Pure Error	0.88	3	0.29		
Cor Total	35.57	12			
**Polysaccharides**					
Model	63.44	5	12..69	9.51	0.0081
Temp	0.12	1	0.12	0.092	0.7722
Hum	17.80	1	17.80	13.34	0.0107
TH	1.36	1	1.36	1.02	0.3512
T^2^	39.61	1	39.61	29.68	0.0016
H^2^	7.18	1	7.18	5.38	0.0595
Residual	8.01	6	1.33		
Lack of fit	7.81	3	2.60	40.08	0.0064
Pure Error	0.19	3	0.065		
Cor Total	83.21	12			

**Table 3 toxins-10-00499-t003:** Experimental ranges and levels of the independent variables.

Variables	Symbols	Range and Levels					Remarks
		−1	−alpha	0	+alpha	+1	
Temperature (°C)	T	22.93	20	30	40	37.03	predicted
		23	20	30	40	37	actual
Humidity (°C)	H	82.2	80	87.5	95	92.8	predicted
		83	80	88	95	93	actual

**Table 4 toxins-10-00499-t004:** HESI-MS/MS parameters of ST.

Compound	Collision energy (kv)	Quantitative Ion (*m*/*z*)	Qualitative Ion (*m*/*z*)
ST	36/25	325 > 281	325 > 310

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
