# Peer review of "Optimization of Storage Conditions of the Medicinal Herb Ilex asprella against the Sterigmatocystin Producer Aspergillus versicolor Using Response Surface Methodology"

_toxins, 2018, doi:10.3390/toxins10120499_

Round 1

Reviewer 1 Report

The present study of contaminant growth, delimited to the specific case of A.Versicolor on I.Asprella is well conceived.

- The paper is well structured and clear.

- Fig.2 needs some work. What is the blue line? Subplots A, B, C should be clearly labeled.

- The model would be easier to interpret if variables were called evocatively T and H, rather than A and B. Due to the choice of parameterization one has to resort to the pictures to understand if models for ST, saponines and polysaccharides are centered around the same values of A and B.

A better parametrization would be  p*(T-T0)^2+q*(H-H0)^2+r(T-T0)(H-H0).

- I wish the conclusion were more mathematically stringent. The main unknowns I can think of are storage time and ST maximum tolerance. Therefore, for example at the prescribed edge values of temperature and humidity, how long can be I asprella stored? I think the conclusions should be reworded in these terms.

Author Response

Response to Reviewer 1 Comments

- Fig.2 needs some work. What is the blue line? Subplots A, B, C should be clearly labeled.

The blue line is the integral line. “A, B, C”  have been added in Fig.2 to label the subplot.

- The model would be easier to interpret if variables were called evocatively T and H, rather than A and B. Due to the choice of parameterization one has to resort to the pictures to understand if models for ST, saponines and polysaccharides are centered around the same values of A and B. A better parametrization would be  p*(T-T0)^2+q*(H-H0)^2+r(T-T0)(H-H0).

Thank you for the suggestion. We had modified the parametrization accordingly.

(1)

ST: Y=-0.034523 * T2 -0.06404* H2+0.015333 *   T *H+0.6822 * T+10.8533* H-471.14569

(2)

Total saponins: Y=0.028044* T2+0.025057 * H2+0.008*   T * H-2.50476 * T-4.81394* H+261.07506

(3)

Polysaccharides: Y =0.05042*T2+0.041459*H2-0.017333*T*H-1.53060*T-7.01683*H+347.41562

Where Y is the predicted response (levels of ST, total saponins and polysaccharides, respectively); T and H represent two independent variables temperature and humidity, respectively.

- I wish the conclusion were more mathematically stringent. The main unknowns I can think of are storage time and ST maximum tolerance. Therefore, for example at the prescribed edge values of temperature and humidity, how long can be I asprella stored? I think the conclusions should be reworded in these terms.

Response 3: As has been described above, levels Temp.30,Hum.87.5% for max point (value) ST for 7 days. If the I. asprella was stored in a suitable temperature and humidity environment for A. versicolor and samples could be stored fewer than seven days. Therefore samples of I. asprella should be conserved under low temperature 26 and humidity 85%, to reduce ST content while maintaining the amounts of active ingredients and ensure the I. asprella could be stored for a longer period of time. This study showed that it was scientific and feasible to use RSD-CCD method to assess the conditions of herb storage, and provided new clues for controlling the pollution of medicinal plants by mycotoxin.

Reviewer 2 Report

I had the opportunity to review the paper tittle "Optimization of Storage Conditions of the Medicinal Herb Ilex asprella Against the Sterigmatocystin Producer Aspergillus versicolor Using Response Surface Methodology ". I was interested in reviewing this manuscript as experiment based on response surface methodology combined central composite design. This is an interesting paper that requires some work before it will be ready to publish. I have some issues that I have commented on individually below.

L12 and L16. In my opinion abbreviations in abstract it is no good idea

L21. Make sure to use keywords that are not included in the tittle

L32. Aspergillus flavus?

L99. Are on the basis of data (Table 1) the second-order response models 1, 2, 3 correct? I’m not sure.

L137-144. The Authors should be more precise in the interpretation of figs 3, 4 and 5. Let’s  consider the ST feature. On the basis of data (Table 1), the second-order response model can calculate values of levels factor A and B for max point (value) ST. Here is: factor A = 29.64oC and factor B = 88.03%. All values below reduce ST content. The same can be calculated for total saponin amount and polysaccharide production. But here will be the minimum points (values). Additional errors: the figures 3, 4 and 5 do not shows star points (Temperature=20 and 40; Humidity= 80 and 95); Figure3 and Table1 - ST value ug/kg or ng/mL?

Table 1. This experiments had 13 runs (tests). Standard plan need only 10 runs. I see 5 tests for center points (A=30 and B=87.5). Why?

Table 2. Are on the basis of data (Table 1) the all ANOVA parameters correct? I have other calculations (values). But the conclusions are similar. I do not know anything about statistical software.

Table 3. Errors in temperature levels values. It should be [-1=22.93; 0=30; +1=37.07]. Star points (alfa) are correct.

Fig1. It should be better quality.

Fig 2. It should be better quality.

Author Response

Response to Reviewer  2 Comments

L12 and L16. In my opinion abbreviations in abstract it is no good idea

Response 1:  We have deleted the abbreviations in abstract.

L21. Make sure to use keywords that are not included in the tittle

Response 2: We have added two more keywords “temperature” and “humidity”.

L32. Aspergillus flavus?

Response 3:   The sentence is improper. We have re-written it.

L99. Are on the basis of data (Table 1) the second-order response models 1, 2, 3 correct? I’m not sure.

 Response 4:  We have recalculated the data using  Design Expert 8.0.6 Analysis Software and corrected the second-order response models 1, 2, 3.

L137-144. The Authors should be more precise in the interpretation of figs 3, 4 and 5. Let’s  consider the ST feature. On the basis of data (Table 1), the second-order response model can calculate values of levels factor A and B for max point (value) ST. Here is: factor A = 29.64oC and factor B = 88.03%. All values below reduce ST content. The same can be calculated for total saponin amount and polysaccharide production. But here will be the minimum points (values). Additional errors: the figures 3, 4 and 5 do not shows star points (Temperature=20 and 40; Humidity= 80 and 95); Figure3 and Table1 - ST value ug/kg or ng/mL?

The paragraph interpreting Figs 3, 4 and 5 has been re-written as follows:

As shown in Figure 3 temperatures and humidity levels ranging from 26 to 34 and 85 to 90%, respectively, significantly increased ST content, indicating that these conditions were optimal for A. versicolor to produce ST. Figure 4 shows that temperature ranging from 26 to 37 and relative humidity of 85-95% significantly reduced the amounts of total saponins. On the basis of data (Table 1), the second-order response model can calculate values of levels Tem.30,Hum.87.5% for max point (value) ST. According to the Table 1,the total saponin  amount was minimum when Temp.30,Hum.87.5% . The content of polysaccharides have a same trend could be seen from three dimensional (3D) surface plots (Figure 5). Therefore, to promote plant quality and safety, storage conditions should be humidity below 85% at a temperature under 26. To ensure optimal chemical composition and good quality for the plant, the recommended storage conditions would be humidity below 85% and temperature below 26.

Additional errors has been re-written as follows:

 The figures 3, 4 and 5 have redisplay star points (Temperature=20 and 40; Humidity= 80 and 95) and Figure3 and Table1 - ST value ug/kg.

Table 1. This experiments had 13 runs (tests). Standard plan need only 10 runs. I see 5 tests for center points (A=30 and B=87.5). Why?

Response 6: According to Design Expert 8.0.6 Analysis, Same tests were random repetitive experiments , mainly to verify the repeatability and accuracy of the experimental results. There were deemed essential.

Table 2. Are on the basis of data (Table 1) the all ANOVA parameters correct? I have other calculations (values). But the conclusions are similar. I do not know anything about statistical software.

 Response 7: I have recalculated the data. The previous error was due to unit conversion, it caused error result ,  but the conclusions are similar.

A short paragraph was added in the section “Materials and Methods 4.7.”, giving the information of the softwares used in this study.

4.7. Statistical Analysis

Stat-Ease software (Design-Expert 8.0.6 version, Stat-Ease Corporation, Minneapolis, MN, USA) was used for the regression analysis of the data, and to plot the response surface graphs. The variability and accuracy of the model were determined according to the regression coefficient (R2) and lack of fit, respectively. Analysis of variance (ANOVA) was performed and the values were considered significant when p < 0.05. The response surface and contour plots of the predicted responses of the model were used to assess interactions between the significant factors. Additionally, numerical optimization was carried out by performing three-dimensional response surface analysis of the independent and dependent variables.

Table 3. Errors in temperature levels values. It should be [-1=22.93; 0=30; +1=37.07]. Star points (alfa) are correct. 

Response 8: Errors have been corrected.

Fig1. It should be better quality.

 Response 9: We have replaced Fig1.with a new one of higher resolution.

Fig 2. It should be better quality.

Response 10: We have replaced Fig 2.with a new one of higher resolution.

Round 2

Reviewer 1 Report

Dear Editor,

I found that a sufficient part of my requirements was fulfilled.

There is a problem: most of the text newly introduced with this revision is plagued by language errors. Most relevantly lines 211-212 and lines 233-234.

Author Response

I found that a sufficient part of my requirements was fulfilled.

Thank you!

There is a problem: most of the text newly introduced with this revision is plagued by language errors. Most relevantly lines 211-212 and lines 233-234.

Sorry for our carelessness. We have revised the first paragraph (lines 211-212) as follows:

As shown in Figure 3, temperatures and humidity levels ranging from 26 to 34and 85 to 90%, respectively, significantly increased ST content, indicating that these conditions were optimal for A. versicolor to produce ST. Based on the data in Table 1, the temperature and humidity for maximum production of ST were calculated to be 30and 88%, respectively. In comparison, increasing temperature ranging from 26 to 37and relative humidity of 85-95% reduced the amounts of total saponins and polysaccharides remarkably (Figure 4 and 5). Therefore, to ensure optimal chemical composition for the herb as well as minimize ST-hazard risk, the recommended storage conditions would be humidity below 85% and temperature below 26.

The conclusion (lines 233-234) was corrected, too. We tried to write the conclusion more mathematically stringent as you suggested last time. A legal limit of 5ppm for food type A and 20ppm for food type B has been set in Czech in 2003. However, it is difficult to draw a conclusion on storage time in this study, considering the inoculated samples are not exact same as real samples. The accumulation of ST still depends on the number of mildews, besides temperature and humidity.

Response surface methodology allows the determination of optimal response based on a few sets of experiments in which all factors vary within a chosen range. Conclusively, the proposed model has shown to be significantly important in statistical terms. This study showed that A. versicolor growth and toxicity were affected by environmental temperature and humidity. The effects of humidity on growth, mycotoxin production and the alteration of effective components were higher than those of temperature. Samples of I. asprella should be conserved under low temperature 26and humidity 85%, to reduce ST content while maintaining the amounts of active ingredients. However, concerning the storage period, no conclusion could be drawn from this study. A long-term experiment with real samples might be necessary to figure out the answer. This study showed that it was scientific and feasible to use RSD-CCD method to assess the conditions of herb storage, and provided new clues for controlling the pollution of medicinal plants by mycotoxin.
